# Investigation of the Deformation Behaviour and Resulting Ply Thicknesses of Multilayered Fibre–MetalFibreM Laminates

**Missam Irani** [1,*], **Moritz Kuhtz** [2], **Mathias Zapf** [1], **Madlen Ullmann** [1], **Andreas Hornig** [2], **Maik Gude** [2] **and Ulrich Prahl** [1]

1. Institute of Metal Forming, Technische Universität Bergakademie Freiberg, 09599 Freiberg, Germany; Mathias.Zapf@imf.tu-freiberg.de (M.Z.); Madlen.Ullmann@imf.tu-freiberg.de (M.U.); Ulrich.Prahl@imf.tu-freiberg.de (U.P.)
2. Institute of Lightweight Engineering and Polymer Technology, Technische Universität Dresden, 01069 Dresden, Germany; moritz.kuhtz@tu-dresden.de (M.K.); andreas.hornig@tu-dresden.de (A.H.); maik.gude@tu-dresden.de (M.G.)
* Correspondence: Missam.Irani@imf.tu-freiberg.de

**Abstract:** Multilayered fibre–metal laminates (FMLs) are composed of metal semifinished products and fibre-reinforced plastics, and benefit from the advantages of both material classes. Light metals in combination with fibre-reinforced thermoplastics are highly suitable for mass production of lightweight structures with good mechanical properties. As the formability of light metal sheets is sometimes limited at room temperature, increasing the process temperature is an appropriate approach to improve formability. However, the melting of thermoplastic materials and resulting loss of stiffness limit the processing temperature. Since single-ply layers have different through-thickness stiffnesses, the forming process changes the ply thickness of the multilayered laminate. In the present study, the deformation behaviour of multilayered FMLs was investigated using a two-dimensional finite-element model assuming plane strain. The thermoelastic-plastic finite-element analysis made investigation of the variation in thickness made possible by incorporating sufficient mesh layers in the thickness direction. The results indicate that a thermoelastic-plastic finite-element model can predict the delamination of plies during deformation, as well as in the final product. Additionally, the predicted changes in thickness of the plies are in good agreement with experimental results when a temperature-dependent friction coefficient is used.

**Keywords:** fibre–metal laminates; finitelf element analysis; plyp thickness; channelc forming

## 1. Introduction

The general public is becoming increasingly mobile. To meet the demands of the automotive industry, novel materials are required to realize highly safe and energy-efficient products. Unfortunately, most materials are not both high-strength (important for safety) and lightweight [1]. However, composites, which are combinations of different materials, have the advantages of good ductility, impact resistance/damage tolerance, high specific stiffness, and corrosion and fatigue resistance [2]. Further improvements can be obtained by optimizing the design of the product, e.g., by varying the layup design, materials, and orientation of the reinforced polymer [3,4].

The consolidation of several materials into a composite is still a major challenge for materials research, due to the complexity of the interactions between the components and the influence of the many parameters associated with the forming processes [5]. Control of spring-in/back, wrinkling, thickness, and even delamination are the main challenges to forming high-performance fibre–metal laminates (FMLs). As a better-developed technology, the deformation modes in the metal-forming process are relatively well-understood. However, the behaviour of FMLs during forming is determined by the deformation mechanisms of metals and polymers, as well as by inter-ply interactions. Understanding and

controlling the flow behaviour of polymer-metal composites remains a major challenge to forming FML components [6]. In this context, modelling the forming processes of FMLs is very important to understand the influence of the process variables, and to optimize them to minimize defects.

Many studies [7–19] have been carried out on the forming of FMLs. Werner et al. [7] and Behrens et al. [8] carried out numerical modelling of a hybrid forming process for three-dimensionally curved FMLs using shell elements for all plies. Dou et al. [11] produced a finite-element model of stamp forming of FMLs, and showed that the properties of the core composite layers strongly influenced the forming behaviour of the FMLs. Sexton et al. [12] also investigated the influence of forming conditions on FMLs, including material properties and their interactions, by finite-element analysis. They used shell elements to model the metal composite. Stamping of FMLs at the thermoforming temperature of the thermoplastic matrix was investigated by Hahn et al. [13]. They established an analytical model to predict the process forces during formation. Wollmann et al. [15] investigated the forming behaviour of FMLs experimentally, analytically, and numerically for cup deep-drawing. In their finite-element analysis, each layer of the sandwich was represented by an individual shell mesh, which interacted with each other through a contact formulation. Dharmalingam et al. [16] used a finite-element method and statistical experiments to determine the influence of forming process parameters. Four node shell elements were used for the FML blank in their simulations. Mosse et al. [17] developed a three-dimensional (3D) finite element analysis of the stamping of a three-layer FML. They used elastic-plastic shell elements for metal plies made of 5005-grade aluminium, and textile elements, to allow for large in-plane shear deformations with no bending stiffness of the middle polymer ply. Rajabi and Kadkhodayan [18] investigated the drawing behaviour of a thermoplastic-based FML consisting of glass-fibre-reinforced polypropylene laminate as the core and aluminium AA1200-O as skin layers. In their study, explicit shell elements were used in a 3D model.

To date, most studies have assumed that FML forming is a mechanical process that occurs at a certain temperature. Furthermore, in most finite-element analyses, polymers were modeled using shell elements; this enabled their modeling as a viscoelastic or viscoplastic material. With the above-mentioned modeling techniques, FML deformation behaviour, as well as defects such as wrinkling and delamination, can be predicted. However, transverse flow cannot be captured using shell or membrane-type elements. Thus, accurate prediction of thickness variation due to polymer flow is not possible.

This study aims to develop a finite element model that predicts not only the defects during and after deformation but also the variation of thickness of polymer plies. Herein, we investigated the channel forming of a multilayered FML composed of Al-5754 aluminium alloy sheets and thermoplastic polyamide 6 reinforced with carbon fibre. This FML was a continuous carbon fibre-reinforced plastic (CFRP) having anisotropic behaviour. The forming process was nonisothermal, and the heat loss caused an inhomogeneous distribution of temperature. A large difference between the thermal properties of the metallic and polymer plies led to still higher temperature distribution inhomogeneity. As the formability of metals and polymers is strongly temperature-dependent, finite-element analysis can provide useful results regarding the flow of metallic and polymer plies. Therefore, deformation of the multilayered FML was studied using a thermoelastic-plastic finite-element model, which made investigation of the variation of thickness possible by incorporating sufficient mesh layers in the thickness direction. Notably, the actual stresses in the CFRP and Al-5754 plies may be different, depending on the viscoplastic behaviour of the polymer [20]. However, the developed thermoelastic-plastic finite-element model predicted the delamination defects, and the variation in thickness of the CFRP and Al-5754 plies, with tolerable error. Due to the geometry of the product, a two-dimensional (2D) finite-element analysis was performed under the plane strain assumption. The predicted results were compared with the experimental data in terms of defect occurrence, as well as the shape and thickness of the plies.

## 2. Materials and Methods

Multilayered FMLs composed of thermoplastic polyamide 6 reinforced with carbon fibre (CFRP) (Celanese CELSTRAN® CFR-TP PA6 CF60-01, Dallas, TX, USA) and Al-5754 aluminium alloy sheets were used in the experiments. Table 1 provides the chemical composition of the Al-5754 used in this study. The plies were bonded to each other using an adhesive layer based on polyolefin, i.e., Cox 391 (nolax AG, Sempach, Switzerland) with a thickness of 0.1 mm. The two lay-ups of multilayered FML used in this study are shown in Figure 1 and their details are given in Table 2. Figure 2 shows the manufacturing steps, including schematic illustrations of the adhesion and forming processes. Due to the different thermal conductivity of the individual layers, laminates were equipped with internal thermocouples in a preliminary study in order to determine a correlation between the external temperature profiles and the temperatures occurring in the laminate for all manufacturing steps.

**Table 1.** Composition of Al-5754 aluminium alloy (wt %).

| Si | Fe | Cu | Mn | Mg | Cr | Zn | Ti | Al |
|----|----|----|----|----|----|----|----|----|
| 0.4 | 0.4 | 0.1 | 0.5 | 3.0 | 0.3 | 0.2 | 0.15 | Balance |

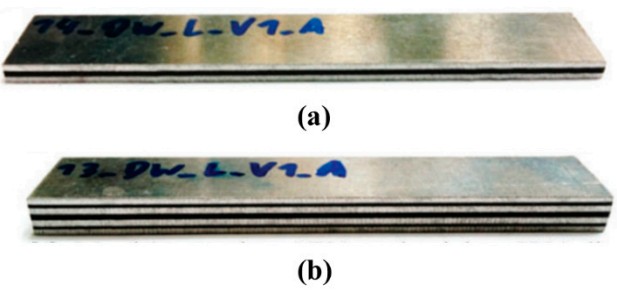

**Figure 1.** Lay-up designs. (**a**) L1 and (**b**) L2.

**Table 2.** Details of the designed lay-ups.

| Lay-Up | Dimensions (mm) | Mesh Size of M Plies in the Simulations (mm) |
|--------|-----------------|----------------------------------------------|
| L1 [M/C]s | [1.0/0.5]s | 0.25 |
| L2 [(M/C)2]s | [(1.0/0.5)2]s | 0.25 |

M: metal (Al-5754 aluminium alloy); C: carbon fibre-reinforced polyamide 6.

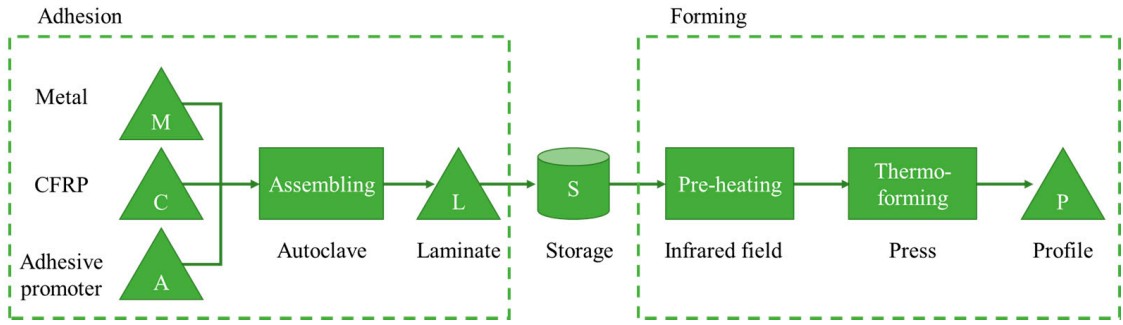

**Figure 2.** Conceptual flowchart of the adhesion and forming process.

In the adhesion step, aluminium alloy sheets, CFRP, and adhesive layers were assembled in an autoclave. Due to the high thickness of the multilayered FML and low through-thickness heat conduction, the stacks were preheated at 100 °C in a vacuum oven (Nabertherm GmbH, TR1050, Lilienthal, Germany) for 2 h. This pre-step also served as a

drying process, since polyamide properties are moisture-dependent. Then, the stacks were consolidated for 1 h at 240 °C and 6 bar. This step was followed by a 1 h cooling step to reduce residual stresses. The consolidated laminates were stored before forming.

Before the deformation step, an infrared field heated the multilayered laminates to 250 °C. The temperature of the laminates was continuously monitored with two pyrometers integrated in the infrared field. No significant temperature loss was measured during the short transport to the press. The dies (Figure 3) were heated to 80 °C and closed when the multilayered FML was inserted. The temperature at the outer surfaces of the workpiece reduced to 95 °C within 20 s due to heat transfer to the dies at a cooling rate of 8 °C/s. The upper die moved at 2 mm/s. The force-controlled press enabled investigation of the influence of a constant pressing force on the deformation behaviour and resulting ply thicknesses. The workpiece was cooled to room temperature after the deformation.

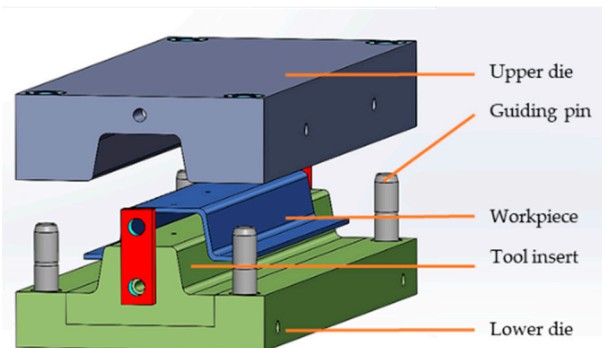

**Figure 3.** Schematic set-up of the forming process.

## 3. Finite-Element Analysis

A 2D thermoelastic-plastic finite-element model was developed assuming plane strain. In order to improve computational efficiency, the dies were modelled as rigid bodies. The finite-element analysis was conducted using a general-purpose metal-forming simulator (Simufact Forming). For mesh generation, plane strain-type quadrilateral elements were used. The mesh sizes used in the various lay-ups are provided in Table 2. To simulate the forming process more accurately, Nolax layers with a thickness of 0.1 mm and mesh size of 0.025 mm were modelled separately to bond the polymer and metal plies in all of the lay-ups (as shown in Figure 4 for L2). The forming load of 75 and 150 kN were applied at the final stroke of the deformation in the simulation of L1 and L2, respectively. These values are the same as the maximum applied load in the experiments. Tangential movement of the layers was restricted by friction, while an adhesive force in the normal direction prevented the layers from separating up to a certain separation stress. No adhesive force above the melting temperature of Nolax (130 °C) was applied. However, 10.0 MPa pressure was applied when the temperature was below 130 °C at the interfaces.

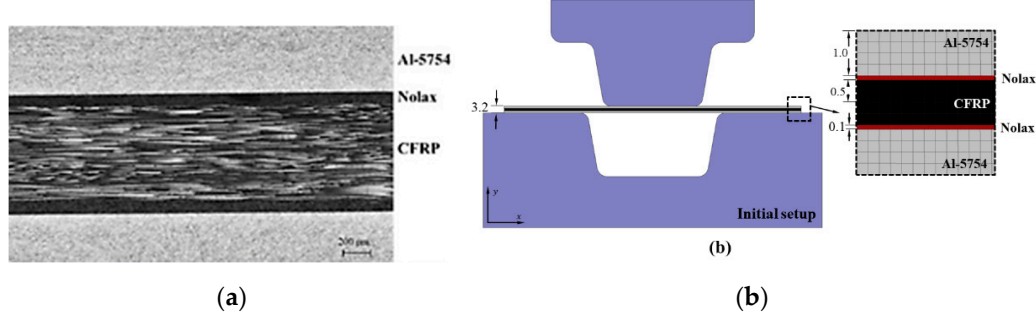

**Figure 4.** (**a**) Micrograph of lay-up L2 and (**b**) finite-element model and meshing system (dimensions in mm).

As the process was nonisothermal, free surfaces were subjected to natural convection and radiation. Heat loss through these free surfaces was calculated by considering both convection and radiation. Newton's law of cooling describes the convective heat transfer on surfaces [21,22]. Moreover, conduction losses needed to be considered from the workpiece to the dies. For the CFRP plies, heat capacity and thermal conductivity of 1700 J/kg·°C and 0.26 W/m·°C were applied, respectively. The thermal properties and elastic modulus of Al-5754 applied to the simulations are given in Table 3.

**Table 3.** Thermal and mechanical properties of Al-5754 aluminium alloy.

| Temperature (°C) | Thermal Conductivity (J/kg·°C) | Heat Capacity (W/m·°C) | Elastic Modulus (GPa) |
|---|---|---|---|
| 50 | 140.0 | 0.93 | 69.0 |
| 100 | 145.0 | 0.96 | 68.0 |
| 150 | 150.0 | 0.975 | 65.0 |
| 200 | 154.0 | 0.99 | 63.0 |
| 250 | 157.0 | 1.00 | 60.0 |

The flow curves of Al-5754 plies were determined in the finite-element analysis via the model developed by Hensel–Spittel according to Equation (1), as follows [23,24]:

$$\sigma = A e^{m_1 T} \varepsilon^{m_2} \dot{\varepsilon}^{m_3} e^{m_4/\varepsilon} \tag{1}$$

where $\sigma$, $\varepsilon$, $\dot{\varepsilon}$ and $T$ are the flow stress, plastic strain, strain rate, and temperature, respectively. $m_1$ to $m_4$ were obtained through compression tests and are given in Table 4. The elastic modulus of CFRP was determined by dynamic mechanical analysis (Figure 5). The flow behaviour of the CFRP plies required for the finite-element analysis was obtained via the material model developed by Wang et al. [25] according to Equation (2), as follows:

$$\frac{d\sigma}{d\varepsilon} = E - E \left( \frac{\sigma}{\sigma^*} \right)^n \tag{2}$$

where $\sigma^*$ and $n$ are the stress coefficient and stress exponent, respectively. The stress exponent, which controls the strain rate strengthening and strain hardening effects of the composite, is strain rate- and temperature-independent. The stress coefficient, on the other hand, varies with both strain rate and temperature, and was linearly extrapolated for the higher temperatures at which the experimental data are not reported. Note that the plane section in which the 2D simulations were developed assuming plane strain was parallel to the fibre direction (Figure 4a). The flow behaviour of the CFRP in the fibre direction was defined in the finite-element analysis by a constant $n$ value of 1.05, and the temperature- and strain rate-dependent stress coefficients provided in Table 5.

**Table 4.** Hensel–Spittel model constants for Al-5754.

| $A$ | $m_1$ | $m_2$ | $m_3$ | $m_4$ |
|---|---|---|---|---|
| 335.92 | −0.00167 | 0.10085 | −0.00058 | 0.0 |

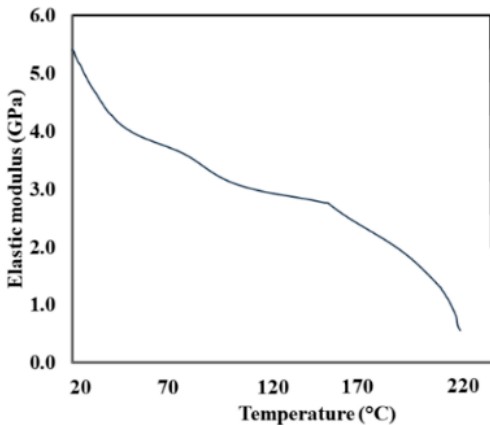

**Figure 5.** Variation in elastic modulus of polyamide 6 as a function of temperature.

**Table 5.** Stress coefficient for CFRP at different strain rates and temperatures.

| Strain Rate (s⁻¹) | Temperature (°C) | Stress Coefficient ($\sigma^*$; MPa) |
|:---:|:---:|:---:|
| $8.5 \times 10^{-4}$ | 21.5 | 118.8 |
| $8.5 \times 10^{-3}$ | | 132.0 |
| $8.5 \times 10^{-2}$ | | 138.6 |
| $8.5 \times 10^{-4}$ | 50 | 87.1 |
| $8.5 \times 10^{-3}$ | | 94.3 |
| $8.5 \times 10^{-2}$ | | 101.6 |
| $8.5 \times 10^{-4}$ | 75 | 70.6 |
| $8.5 \times 10^{-3}$ | | 77.2 |
| $8.5 \times 10^{-2}$ | | 79.6 |
| $8.5 \times 10^{-4}$ | 100 | 58.7 |
| $8.5 \times 10^{-3}$ | | 62.9 |
| $8.5 \times 10^{-2}$ | | 64.8 |
| $8.5 \times 10^{-4}$ | 150 | 16.6 |
| $8.5 \times 10^{-3}$ | | 14.9 |
| $8.5 \times 10^{-2}$ | | 13.1 |

## 4. Results and Discussion

Interfacial behaviour, especially inter-ply slip, plays an important role in finite-element analysis of layered composites. Some studies [13] assumed no slippage between the metal and polymer plies, so the plies were tied together in the model. When interfacial slippage is considered, Coulomb's friction law is usually applied and the friction coefficient is required as input for the simulations [8]. The friction coefficient applied in the simulations influences the predicted forming loads and determines the volume of the squeezed-out polymer and thicknesses of the plies (Figure 6). Notably, when a friction coefficient of 0.2 was used in our analysis, laying against the lower die surface was incomplete and less polymer accumulation at the head radius of the hat profile channel was predicted (Figure 6c). This behaviour was due to the reduction of squeezed-out polymer at higher friction coefficients. In the current finite-element analysis, a hybrid contact model was used between the cohesive layers of Nolax and the plies of aluminium alloy and polymer. Inter-ply free slippage was permitted when the temperature of the interface exceeded the Nolax melting point, and when the applied stress exceeded the solid-state bonding strength of the plies. The melting point of Nolax and measured bonding strength were 130 °C and 10.0 MPa, respectively.

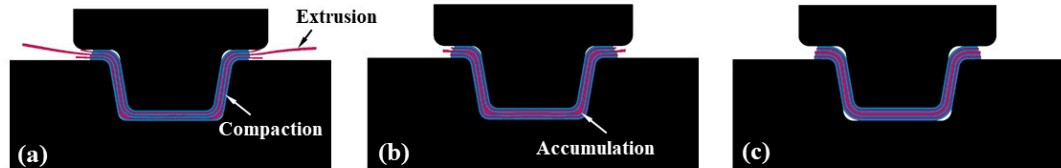

**Figure 6.** Influence of the Coulomb coefficient factor on inter-ply slippage. (**a**) μ = 0.01, (**b**) μ = 0.1, and (**c**) μ = 0.2.

The predicted cross-section of the hat profile channel for lay-up L2 is compared with experimental data in Figure 7. To obtain the appropriate friction coefficient, values of 0.05, 0.1, and 0.2 were applied between the aluminium and polymer plies. The predicted cross-section of the hat profile was compared with the experimental data at a forming load of 10.0 kN, at which point the profile was not fully formed. Figure 7a–c shows that the α angle in the predicted cross-section varied when different friction coefficients were applied. Comparison of the predicted and measured α indicates that the cross-section of the produced hat profile channel was very similar to the cross-section predicted using the coefficient factor of 0.1. The friction coefficient between the dies and workpiece was fixed at 0.1 in all simulations. Thus, the inter-ply friction coefficient, and the friction coefficient between the dies and workpiece, were set at 0.1 in subsequent simulations.

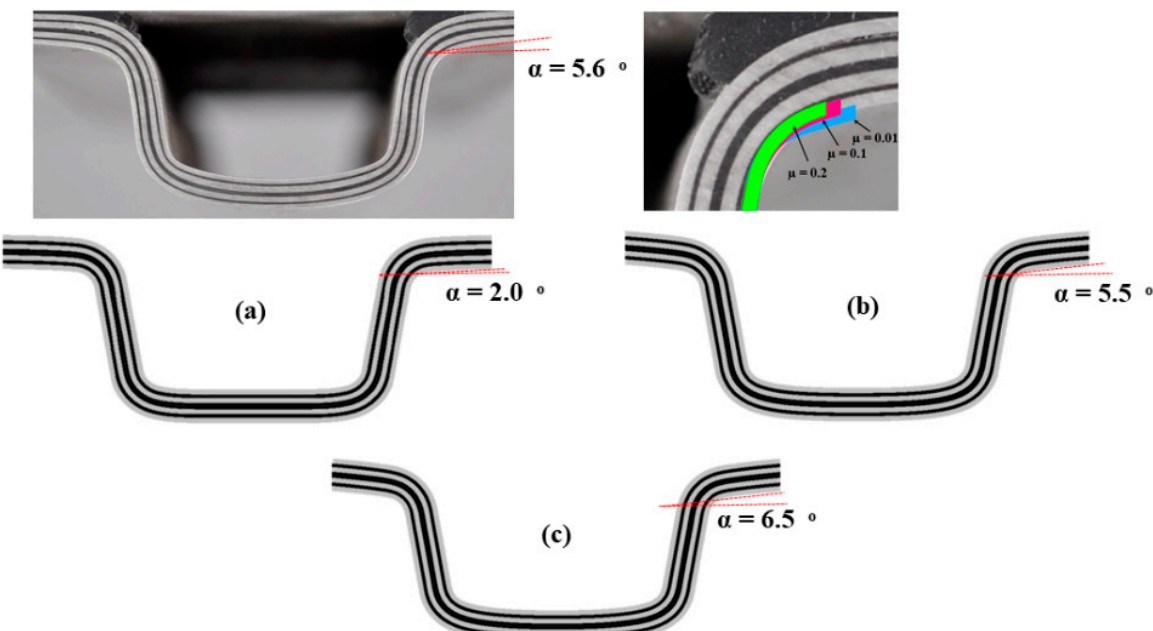

**Figure 7.** Influence of the inter-ply Coulomb coefficient on the α angle. (**a**) μ = 0.01, (**b**) μ = 0.1, and (**c**) μ = 0.2.

The temperature distribution at the final deformation stroke of various lay-ups is illustrated in Figure 8. A constant temperature of 80 °C was assumed for the dies during the analysis. To reduce the calculation time, the dies were assumed to be rigid without heat conduction. As seen in lay-up L2, the low thermal conductivity of the CFRP plies and Nolax cohesive layers reduced heat loss to the aluminium plies located on the surface, and to the dies. Thus, the temperature of the aluminium and CFRP core plies was not greatly reduced even after 20 s of contact with the dies. However, the temperature of the plies was more reduced in L1 due to cooling of both the top and bottom aluminium plies, which were in contact with the dies. The influence of temperature on the flow behaviour of the CFRP plies is also evident in Figure 8. In lay-up L2, a higher amount of CFRP squeezed out on the lateral sides. It was assumed that the CFRP plies could flow without any difficulty at the interface of the plies due to the high temperature. In other words, at high temperatures, the Nolax adhesive layers would not prevent the plies from sliding at the interfaces. Thus,

a higher amount of squeezed-out CFRP in the middle ply of L2 and a greater reduction in ply thickness were expected.

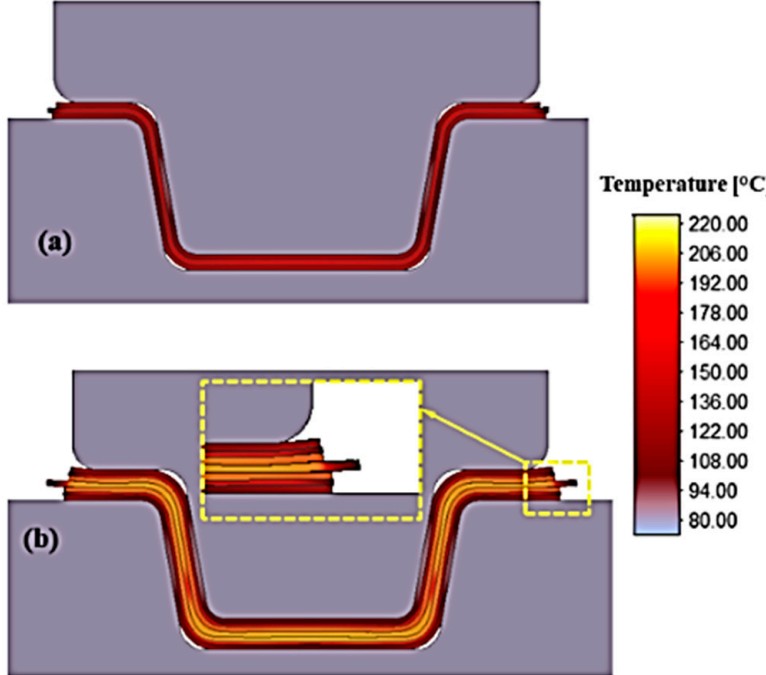

**Figure 8.** Temperature distribution at the final deformation stroke. (**a**) L1 and (**b**) L2.

Delamination, extrusion, compaction, and accumulation of the polymer plies are the main defects that occur during the FLM forming process (Figure 6). To investigate the capability of the developed finite-element analysis to predict delamination, the predicted defects in the cross-section of L1 and L2 were compared with the experimental data (Figure 9). The location of the delamination was well-predicted but the size of the predicted crack at the interfaces was underestimated. Temperature dependency of the bonding strength of Nolax could be one explanation for this, because the bonding strength in the present finite-element analysis was assumed to be constant at 10 MPa. This value was the bonding strength at room temperature, but the predicted temperature at the interfaces of the Al5754 and CFRP plies exceeded 100 °C and was particularly high at the interfaces of the core plies. The cross-section of the semi-formed channel with lay-up L2 at the forming load of 10 kN is illustrated in Figure 9b; delamination occurred between the metal and polymer plies at the flange corner. The simulation also predicted delamination at the same region of the flange under the same forming load (Figure 9b, left). Figure 10 shows that delamination at the web of the hat profile was predicted for lay-up L2 in the middle stages of the forming process. Inter-ply slip caused delamination in this area due to insufficient compressive stress and the high temperature, which reduced the bonding strength of the plies. However, such defects are not usually observed in this region of the final products because they are re-glued via the high compressive stress applied on the web of the hat profile during the final stages of channel forming.

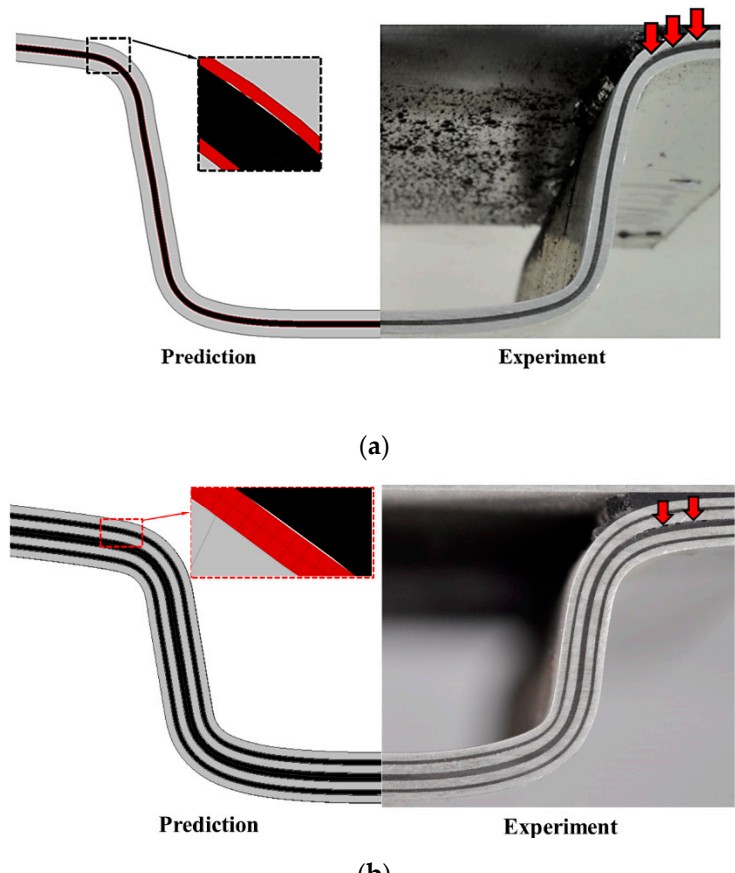

**Figure 9.** Delamination at the metal–polymer interface. (**a**) Final product with L1 and (**b**) semi-deformed profile with L2 at the forming load of 10 kN.

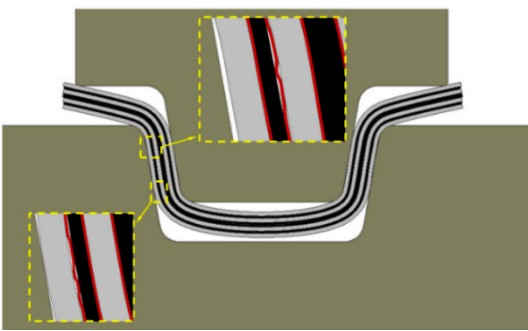

**Figure 10.** Predicted delamination at the metal–polymer interface in a semi-deformed profile with L2 in the middle forming stages.

Figure 11a,b compares the predicted cross-sections of lay-ups L1 and L2 with the experimental findings at the head radius of the hat profile. A reduction in CFRP ply thickness is evident at the flange and head radius for both the predicted and experimental cases. To quantitatively validate the predicted thickness, those of the Al-5754 and CFRP plies were compared with the measured values at positions P1–P3 on the head radius (Table 6). The ply labels (CFRP, CFRP1, CFRP2, and CFRP3) and measurement positions are indicated in Figure 11. Figure 12a shows that the predicted thicknesses of CFRP1 and CFRP3 were in good agreement with the measured values. The maximum calculated error between the predicted and measured values was 5.5% for the plies listed in Table 5. However, the predicted thicknesses for CFRP (L1) and CFRP2 (L2) were quite different from the measured values, with a maximum error of 27.6% for CFRP2 at the P2 position. Note

that the thicknesses reported in Table 6 were predicted using finite-element analysis with a constant Coulomb friction coefficient of 0.1 at all interfaces, while the friction coefficient for polymers is temperature-dependent. Friction patterns as functions of temperature were reviewed by Myshkin et al. [26].

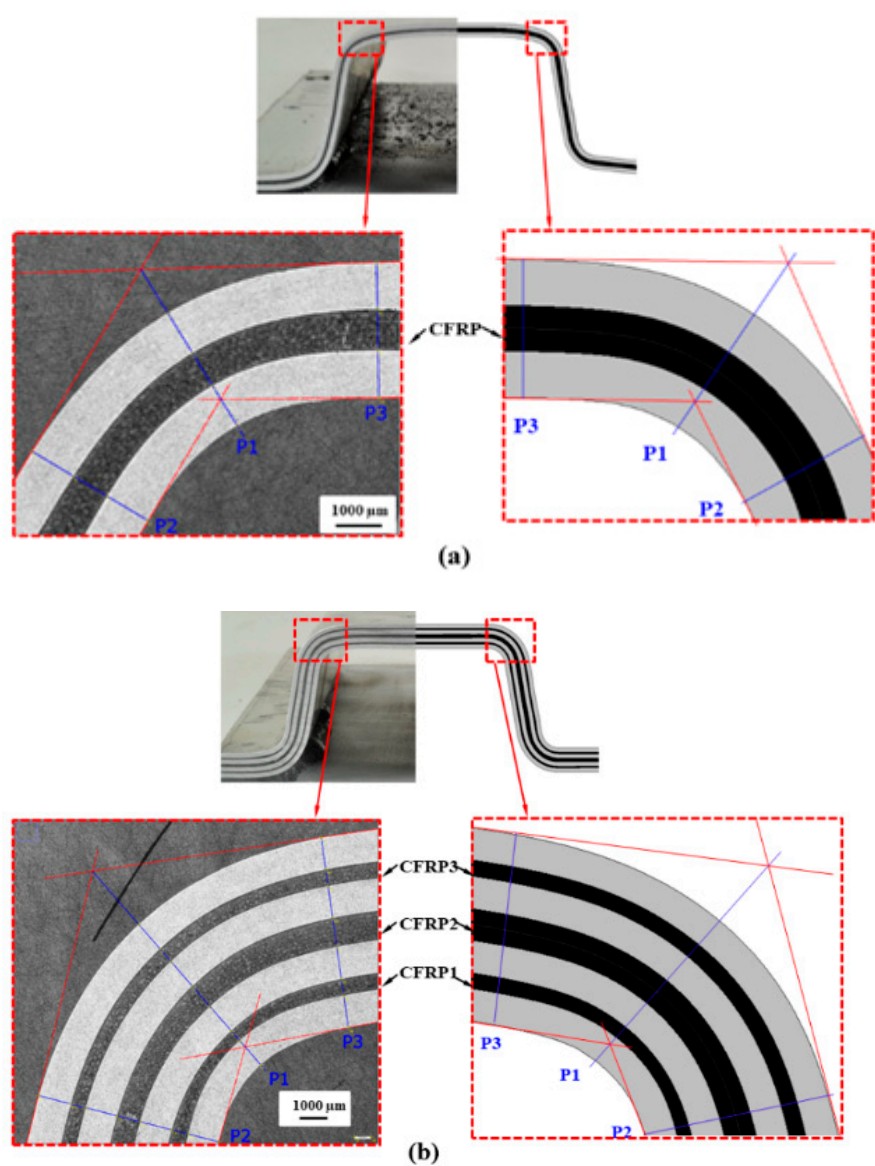

**Figure 11.** Experimental and predicted cross-sections of lay-ups (**a**) L1 and (**b**) L2.

**Table 6.** Measured and predicted thickness of the deformed multilayered FML (constant friction coefficient).

| Lay-Up | Ply | Thickness at P1 (micron) | | | Thickness at P2 (micron) | | | Thickness at P3 (micron) | | |
|---|---|---|---|---|---|---|---|---|---|---|
| | | Measured | Predicted | Error (%) | Measured | Predicted | Error (%) | Measured | Predicted | Error (%) |
| L1 | CFRP | 822 | 971 | 18.1 | 832 | 955 | 14.8 | 881 | 970 | 10.0 |
| | CFRP1 | 430 | 454 | 5.5 | 440 | 445 | 1.1 | 514 | 513 | 0.4 |
| L2 | CFRP2 | 832 | 960 | 15.3 | 737 | 940 | 27.6 | 903 | 1002 | 11.0 |
| | CFRP3 | 509 | 506 | 0.6 | 472 | 495 | 4.7 | 530 | 537 | 1.4 |

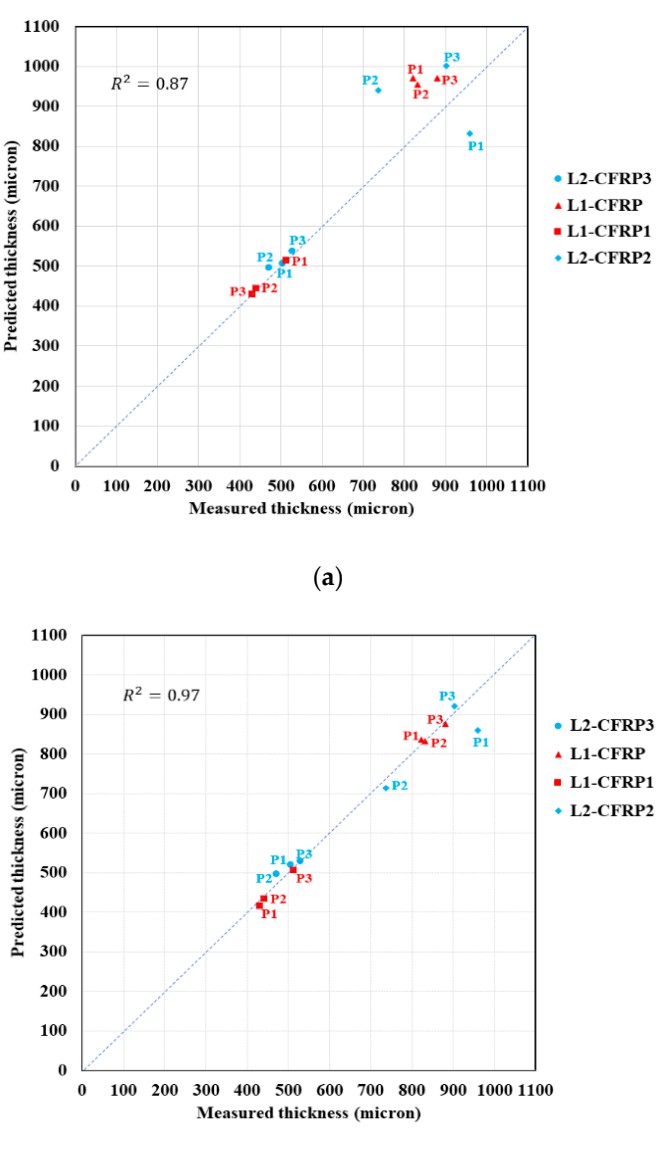

**Figure 12.** Measured and predicted thickness of CFRP plies at the head radius. (**a**) Constant friction coefficient. (**b**) Temperature-dependent friction coefficient.

To improve the finite-element analysis, we applied a linear temperature-dependent friction coefficient to the model in which the friction coefficient at 150 °C and 210 °C were assumed to be 0.1 and 0.05, respectively. Table 7 compares the predicted thickness after applying a temperature-dependent friction coefficient to the experimental data; the maximum error for the predicted thicknesses of CFRP (L1) and CFRP2 (L2) decreased from 27.6% to 3.3%. The temperature at the interfaces of the middle plies was higher, which led to a lower friction coefficient and easier slippage. Thus, the amount of squeezed-out polymer increased, and the predicted thickness of the plies decreased accordingly. Figure 12b shows that the finite-element analysis improved by implementing a temperature-dependent friction coefficient that predicted the thickness changes after deformation with tolerable error and an increase in the value of $R^2$ from 0.87 to 0.97 when a temperature-dependent friction coefficient is implemented.

**Table 7.** Measured and predicted thickness of the deformed multilayered FML (temperature-dependent friction coefficient).

| Lay-Up | Ply | Thickness at P1 (micron) | | | Thickness at P2 (micron) | | | Thickness at P3 (micron) | | |
|---|---|---|---|---|---|---|---|---|---|---|
| | | Measured | Predicted | Error (%) | Measured | Predicted | Error (%) | Measured | Predicted | Error (%) |
| L1 | CFRP | 822 | 836 | 1.7 | 832 | 833 | 0.1 | 881 | 875 | 0.7 |
| | CFRP1 | 430 | 416 | 3.2 | 440 | 435 | 1.1 | 514 | 505 | 1.7 |
| L2 | CFRP2 | 832 | 860 | 3.3 | 737 | 714 | 3.1 | 903 | 920 | 1.9 |
| | CFRP3 | 509 | 520 | 2.1 | 472 | 495 | 4.8 | 530 | 528 | 0.4 |

## 5. Conclusions

The deformation behaviour of a multilayered FML during production of a hat profile channel was investigated using thermoelastic-plastic finite-element analysis. Due to the product geometry, a 2D finite-element analysis was used under the plane strain assumption. Two versions of the finite-element model were developed and validated using two different lay-up designs.

The results revealed that a thermoelastic-plastic finite element analysis, which implemented a constant friction coefficient, did not show acceptable accuracy in its predictions of ply thickness. However, the second model applied a linearly temperature-dependent friction factor at the metal–polymer interface and predicted thickness variation at the head radius of the hat profile with tolerable error. The developed model could also predict delamination at the flange radius of the hat profile channel, where this defect formed in the produced hat profiles. The model with a temperature-dependent friction coefficient is suitable for the design and development of multilayered FML components focusing on the prediction of defects and thickness variation. This model could accurately predict not only the occurrence of defects (such as delamination) during deformation but also the variation in ply thickness.

**Author Contributions:** Conceptualization, M.I., M.K., M.Z., M.U., A.H., M.G and U.P.; methodology, M.I., M.K., M.Z., M.U. and A.H.; software, M.I., M.K., M.Z., M.U. and A.H.; validation, M.I., M.K., M.Z., M.U. and A.H.; formal analysis, M.I., M.K., M.Z., M.U. and A.H.; investigation, M.I., M.K., M.Z., M.U. and A.H.; re-sources, M.G and U.P.; data curation, M.I., M.K., M.Z., M.U. and A.H.; writing-original draft preparation, M.I., M.K. and M.Z.; writing-review and editing, M.U., A.H., M.G. and U.P.; visualization, M.I., M.K., M.Z., M.U., A.H.; supervision, M.G. and U.P.; project administration, M.G. and U.P.; funding acquisition, M.G. and U.P. All authors have read and agreed to the published version of the manuscript.

**Funding:** The authors gratefully acknowledge the financial support for this research provided by the European Union (European Regional Development Fund) and the Free State of Saxony under grant agreement no. 100285086 ("hybcrash") and grant agreement no. 100383375 ("dahlia").

**Institutional Review Board Statement:** Not applicable.

**Informed Consent Statement:** Not applicable.

**Data Availability Statement:** Data available on request due to restrictions e.g., privacy or ethical.

**Conflicts of Interest:** The authors declare no conflict of interest.

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
