# Peer review of "Investigation of the Deformation Behaviour and Resulting Ply Thicknesses of Multilayered Fibre–Metal Laminates"

_jcs, doi:10.3390/jcs5070176_

Round 1
Reviewer 1 Report
The researchers have done work about “Investigation of the Deformation Behaviour and Resulting Ply Thicknesses of Multilayered Fiber–metal Laminates”. However, there are fewclarifications required in the manuscript before publication.
• During adhesion process in autoclave, are you following any standards or trial-and-error method for heating-consolidation-cooling for these material combinations. If you have any reference or standards kindly cite. • Have you mentioned the applied pressure value for forming process? If yes, the same pressure considered for analytical models? • Table 3 kindly remove the repetition of temperature values 50 & 100. • In page no 9, line no 271-273, “To improve the finite-element analysis, we applied a linear temperature-dependent friction coefficient to the model in which the friction coefficient at 150 °C and 210 °C were assumed to be 0.1 and 0.05, respectively.” At this temperature what is the role of Nolax? • Provide the clear images for Figure 9. • It will be good if you able to add micrographs in order to find clear identification of defects and could compare with FEA model. • The recent references were considered for this less than 50% (publications from last 5 years). It is recommended to authors to add more recent relevant references.1. Effect of process parameters on polyamide-6 carbon fibre prepreg laminated by IR-assisted automated fibre placement
C Venkatesan, R Velu, N Vaheed, F Raspall, TE Tay, A SilvaAuthor Response
We would like to thank the reviewers for the valuable comments and very helpful suggestions. We carefully revised the manuscript according to the reviewers’ advice. Also, we addressed the reviewers’ comments point by point. The revision made in the manuscript could be tracked by track changes option in office-word software.
Reviewer 1:
During adhesion process in autoclave, are you following any standards or trial-and-error method for heating-consolidation-cooling for these material combinations. If you have any reference or standards kindly cite.
The suitable temperature profile for the bonded joint was taken from the manufacturer's specifications. Preliminary studies were conducted to determine the time-dependent temperature behaviour. The related information has been added (line 105-109 and 123-124).
Have you mentioned the applied pressure value for forming process? If yes, the same pressure considered for analytical models?
Thank you for your remind. Please check Line 141-143. The required information is added.
Table 3 kindly remove the repetition of temperature values 50 & 100.
The Table is modified.
In page no 9, line no 271-273, “To improve the finite-element analysis, we applied a linear temperature-dependent friction coefficient to the model in which the friction coefficient at 150 °C and 210 °C were assumed to be 0.1 and 0.05, respectively.” At this temperature what is the role of Nolax?
At the determined temperature range, the Nolax does not play the adhesive role but it remains between the plies. Thus what is influencing the slippage of interfaces is friction. However, when the temperature reduces in some regions (please see Figure 8) the sticking role of Nolax would be recovered. This is stated in Lines 145-147
Provide the clear images for Figure 9. • It will be good if you able to add micrographs in order to find clear identification of defects and could compare with FEA model.
We tried to improve the quality of the photos. Unfortunately, we can not provide micrographs in a short period of time.
The recent references were considered for this less than 50% (publications from last 5 years). It is recommended to authors to add more recent relevant references.1. Effect of process parameters on polyamide-6 carbon fibre prepreg laminated by IR-assisted automated fibre placement
C Venkatesan, R Velu, N Vaheed, F Raspall, TE Tay, A Silva
List of References are modified using 3 new references including the above-mentioned paper.
*** The English in this document has been checked by at least two professional editors, both native speakers of English. For a certificate, please see: http://www.textcheck.com/certificate/Y1OQsm
Reviewer 2 Report
There are some weaknesses through the manuscript which need improvement. Therefore, the submitted manuscript cannot be accepted for publication in this form, but it has a chance of acceptance after a major revision. My comments and suggestions are as follows:
1- Abstract gives information on the main feature of the performed study, but some details about the conducted finite element model must be added.
2- Authors must clarify necessity of the performed research. Aims and objectives of the study must be clearly mentioned in introduction.
3- The literature study must be enriched. In this respect, authors must read and refer to the following papers: (a) CFRP: https://doi.org/10.1016/j.euromechsol.2019.103821 (b) Finite element: https://doi.org/10.1016/j.polymertesting.2019.106111
4- It would be nice, if authors could present the schematic figure (Fig. 3) in a high resolution.
5- Standard deviation is the presented curves must be discussed. Details of measured values (mentioned in Tables) must be presented.
6- In its language layer, the manuscript should be considered for English language editing. There are sentences which have to be rewritten.
7- The conclusion must be more than just a summary of the manuscript. List of references must be updated based on the proposed papers. Please provide all changes by red color in the revised version.
Author Response
We would like to thank the reviewers for the valuable comments and very helpful suggestions. We carefully revised the manuscript according to the reviewers’ advice. Also, we addressed the reviewers’ comments point by point. The revision made in the manuscript could be tracked by track changes option in office-word software.
Reviewer 2:
Abstract gives information on the main feature of the performed study, but some details about the conducted finite element model must be added.
The abstract is modifed giving more details of the finite element analysis.
Authors must clarify necessity of the performed research. Aims and objectives of the study must be clearly mentioned in introduction.
The aim of the study is clearly stated by adding lines 77-78
The literature study must be enriched. In this respect, authors must read and refer to the following papers: (a) CFRP: https://doi.org/10.1016/j.euromechsol.2019.103821 (b) Finite element: https://doi.org/10.1016/j.polymertesting.2019.106111
The above mentioned papers are now listed in the references.
It would be nice, if authors could present the schematic figure (Fig. 3) in a high resolution.
New image has been provided.
Standard deviation is the presented curves must be discussed. Details of measured values (mentioned in Tables) must be presented.
The graphs in Figure 12 have been replaced by new graphs which give the R-squared values.
In its language layer, the manuscript should be considered for English language editing. There are sentences which have to be rewritten.
The English in this document has been checked by at least two professional editors, both native speakers of English. For a certificate, please see: http://www.textcheck.com/certificate/Y1OQsm
The conclusion must be more than just a summary of the manuscript. List of references must be updated based on the proposed papers. Please provide all changes by red color in the revised version.
We tried to modify the structure of the conclusion chapter.

Reviewer 3 Report
The authors present a useful study on the deformation of multilayered-metal laminates. The following questions were generated from the review of this 1st version:
- How’s the temperature measured in the experimental set-up?
- How are the dies modelled in the simulations? Fully rigid?
- At the end of page 6, there’s a figure without caption, please be careful with these details.
- Can Figure 7 be enhanced with an overlapping placing of the different results from the different coefficients used? This will aid the comparison.
- Lines 209 and 229, are they part of the caption of a figure or the paragraph. Again careless!
- Can the authors provide a quantitative comparison of the prediction and experimental results of the radii at the curvatures?
Author Response
We would like to thank the reviewers for the valuable comments and very helpful suggestions. We carefully revised the manuscript according to the reviewers’ advice. Also, we addressed the reviewers’ comments point by point. The revision made in the manuscript could be tracked by track changes option in office-word software.
Reviewer 3:
How’s the temperature measured in the experimental set-up?
Preliminary studies were conducted to determine the time-dependent temperature behaviour. A corresponding paragraph was added (line 105-109 and 123-124).
How are the dies modelled in the simulations? Fully rigid?
The information about the simulations of die is added. Please see line 135.
At the end of page 6, there’s a figure without caption, please be careful with these details.
The problem was due to preparing the paper in the requested format by the journal. It is solved now.
Can Figure 7 be enhanced with an overlapping placing of the different results from the different coefficients used? This will aid the comparison.
The figure is modified.
Lines 209 and 229, are they part of the caption of a figure or the paragraph. Again careless!
Thank you for your remind. The problem was due to preparing the paper in the requested format by the journal. It is solved now.
Can the authors provide a quantitative comparison of the prediction and experimental results of the radii at the curvatures?
Unfortunately, the radii cannot be specified because they do not adjust uniformly as can be seen in the illustration. Therefore, for the comparison between simulation and experiment, the representation as in Figure 9 was chosen.
*** The English in this document has been checked by at least two professional editors, both native speakers of English. For a certificate, please see: http://www.textcheck.com/certificate/Y1OQsm

Round 2
Reviewer 2 Report
The paper has been improved and corresponding modifications have been conducted. In my opinion, the current version can be considered for publication.